# MULTI-SAMPLE DROPOUT FOR ACCELERATED TRAINING AND BETTER GENERALIZATION

## ABSTRACT

Dropout is a simple but efficient regularization technique for achieving better generalization of deep neural networks (DNNs); hence it is widely used in tasks based on DNNs. During training, dropout randomly discards a portion of the neurons to avoid overfitting. This paper presents an enhanced dropout technique, which we call multi-sample dropout, for both accelerating training and improving generalization over the original dropout. The original dropout creates a randomly selected subset (called a dropout sample) from the input in each training iteration while the multi-sample dropout creates multiple dropout samples. The loss is calculated for each sample, and then the sample losses are averaged to obtain the final loss. This technique can be easily implemented without implementing a new operator by duplicating a part of the network after the dropout layer while sharing the weights among the duplicated fully connected layers. Experimental results showed that multi-sample dropout significantly accelerates training by reducing the number of iterations until convergence for image classification tasks using the ILSVRC 2012 (ImageNet), CIFAR-10, CIFAR-100, and SVHN datasets. Multi-sample dropout does not significantly increase computation cost per iteration for deep convolutional networks because most of the computation time is consumed in the convolution layers before the dropout layer, which are not duplicated. Experiments also showed that networks trained using multi-sample dropout achieved lower error rates and losses for both the training set and validation set.

## 1 INTRODUCTION

Dropout (Hinton et al. (2012)) is one of the key regularization techniques for improving the generalization of deep neural networks (DNNs). Because of its simplicity and efficiency, the original dropout and various similar techniques are widely used to train neural networks for various tasks. The use of dropout prevents the trained network from overfitting to the training data by randomly discarding (i.e., "dropping") 50% of the neurons at each training iteration. As a result, the neurons cannot depend on each other, and the trained network achieves better generalization. During inference, neurons are not discarded, so all information is preserved; instead, each outgoing value is multiplied by 0.5 to make the average value consistent with the training time. The network used for inference can be viewed as an ensemble of many sub-networks randomly created during training. The success of dropout inspired the development of many techniques using various ways for selecting information to discard. For example, DropConnect (Wan et al. (2013)) discards a portion of the connections between neurons randomly selected during training instead of randomly discarding neurons.

This paper reports *multi-sample dropout*, a dropout technique extended in a different way especially for deep convolutional neural networks (CNNs). The original dropout creates a randomly selected subset (a *dropout sample*) from the input during training. The proposed multi-sample dropout creates multiple dropout samples. The loss is calculated for each sample, and then the sample losses are averaged to obtain the final loss used for back propagation. By calculating losses for $M$ dropout samples and ensembling them, network parameters are updated to achieve smaller loss with any of these samples. This is similar to performing $M$ training repetitions for each input image in the same minibatch. Therefore, it significantly reduces the number of iterations needed for training. We observed that multi-sample dropout also improved accuracy of the trained network; it can reduce the gradient noises caused by dropout. We do not discard neurons in the inference, as with the original

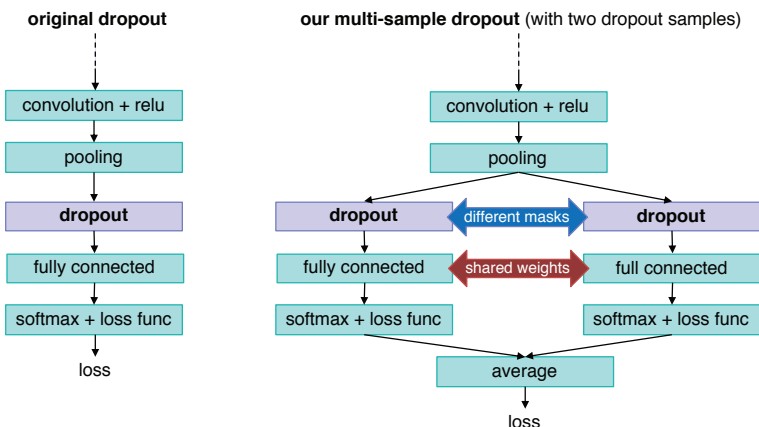

Figure 1: Overview of original dropout and our multi-sample dropout.

dropout. Experiments demonstrated that it achieves smaller losses and errors for both the training set and validation set on image classification tasks.

As far as the authors know, no other dropout technique uses a similar approach to accelerate training. In CNNs, dropout is typically applied to layers near the end of the network. VGG16 (Simonyan & Zisserman (2014)), for example, uses dropout for 2 fully connected layers following 13 convolution layers. Because the execution time for the fully connected layers is much shorter than that for the convolution layers, duplicating the fully connected layers for each of multiple dropout samples does not significantly increase the total execution time per iteration. Experiments using the ImageNet, CIFAR-10, CIFAR-100, and Street View House Numbers (SVHN) datasets showed that, with an increasing number of dropout samples created at each iteration, the improvements obtained (reduced number of iterations needed for training and higher accuracy) became more significant at the expense of a longer execution time per iteration and greater memory consumption. Consideration of the reduced number of iterations along with the increased time per iteration revealed that the total training time was the shortest with a moderate number of dropout samples, such as 8.

Multi-sample dropout can be easily implemented on deep learning frameworks without adding a new operator by duplicating a part of the network after the dropout layer while sharing the weights among the fully connected layers duplicated for each dropout sample.

The main contribution of this paper is multi-sample dropout, a new regularization technique for accelerating the training of deep neural networks compared to the original dropout. Evaluation of multi-sample dropout on image classification tasks demonstrated that it increases accuracy for both the training and validation sets as well as accelerating the training.

## 2 MULTI-SAMPLE DROPOUT

### 2.1 OVERVIEW

This section describes the multi-sample dropout technique. The basic idea is quite simple: create multiple dropout samples instead of only one. Figure 1 depicts an easy way to implement multi-sample dropout (with two dropout samples) using an existing deep learning framework with only common operators. The dropout layer and several layers after the dropout are duplicated for each dropout sample; in the figure, the "dropout," "fully connected," and "softmax + loss func" layers are duplicated. Different masks are used for each dropout sample in the dropout layer so that a different subset of neurons is used for each dropout sample. In contrast, the parameters (i.e., connection weights) are shared between the duplicated fully connected layers. The loss is computed for each dropout sample using the same loss function, e.g., cross entropy, and the final loss value is obtained by averaging the loss values for all dropout samples. This final loss value is used as the objective function for optimization during training. We select the class label as the prediction based on the average of outputs from the last fully connected layer. Although a configuration with two dropout

samples is shown in Figure 1, multi-sample dropout can be configured to use any number of dropout samples. The original dropout can be seen as a special case of multi-sample dropout where the number of samples is set to one.

During inference, Neurons are not discarded as is done in the original dropout. The loss can be calculated using only one dropout sample because the dropout samples become identical at the inference time if we do not drop any neurons. Hence, We always use only one dropout sample at inference regardless of the training method.

When dropout is applied to a layer near the end of the network, the additional execution time due to the duplicated operations is not significant; this characteristic makes multi-sample dropout more suitable for deep CNNs. We can apply multi-sample dropout for shallow networks, such as the multilayer perceptron. As we show an example in Appendix, multi-sample dropout reduces the number of iterations for training, but the costs of the increased execution time per iteration may surpass the benefits.

If the network includes multiple dropout layers, we can apply multi-sample dropout at any of these dropout layers. Multi sampling at an earlier dropout layer may increase diversity among dropout samples and increase the benefits in trade for the higher additional costs due to more duplicated layers. For example, WideResNet (Zagoruyko & Komodakis (2016)) repeatedly executes dropout since it employs residual blocks which include dropout layers. However, we do not need to duplicate the entire network for applying multi-sample dropout to WideResNet; we can apply the multi sampling at the last residual block for example.

## 2.2 Why multi-sample dropout accelerates training

Intuitively, the effect of multi-sample dropout with $M$ dropout samples is similar to that of enlarging the size of a minibatch $M$ times by duplicating each sample in the minibatch $M$ times. For example, if a minibatch consists of two data samples $\langle A, B \rangle$, training a network by using multi-sample dropout with two dropout samples closely corresponds to training a network by using the original dropout and a minibatch of $\langle A, A, B, B \rangle$. Using a larger minibatch size with duplicated samples may not make sense because it increases the computation time $M$ times. In contrast, multi-sample dropout can enjoy similar gains without a huge increase in computation cost for deep CNNs because it duplicates only the operations after dropout. Because of the non-linearity of the activation functions, the original dropout with duplicated samples and multi-sample dropout do not give exactly the same results. However, similar acceleration was observed in the training in terms of the number of iterations, as shown by the experimental results.

## 2.3 Other sources of diversity among samples

The key to faster training with multi-sample dropout is the diversity among dropout samples; if there is no diversity, the multi-sampling technique gives no gain and simply wastes computation resources. Although dropout is one of the best sources of diversity, the multi-sampling technique can be used with other sources of diversity. For example, variants of dropout, such as DropConnect, can be enhanced by using the multi-sampling technique.

To demonstrate that benefits can be obtained from other sources of diversity, two additional diversity-creation techniques are employed in this paper: 1) horizontal flipping and 2) zero padding at a pooling layer. As shown in Appendix, improvements due to these additional sources of diversity are visible but much less significant than the improvements from dropout. Such results show that dropout is an ideal source of diversity to use in our multi-sampling technique. In this paper, these additional sources of diversity are used only in the training phase, but not in the inference.

## 3 Experimental Results

### 3.1 Implementation

This section describes the effects of using the multi-sample dropout for various image classification tasks using the ImageNet, CIFAR-10, CIFAR-100, and SVHN datasets. For the ImageNet dataset, as well as for the full dataset with 1,000 classes, a reduced dataset with only the first 100 classes

was tested (ImageNet-100). For most of the experiments, we use eight as the number of dropout samples, which generally gives good tradeoff between benefits and additional cost. For the CIFAR-10, CIFAR-100, and SVHN datasets, an 8-layer network with six convolutional layers and batch normalization (Ioffe & Szegedy (2015)) followed by two fully connected layers with dropout was used (see appendix for detail). This network executes dropout twice with dropout ratios of 40% and 30%, which are tuned for the original dropout but are used here for all cases unless otherwise specified. The same network architecture except for the number of neurons in the output layer was used for the CIFAR-10, SVHN (10 output neurons), and CIFAR-100 (100 output neurons) datasets. The network was trained using the Adam optimizer (Kingma & Ba (2014)) with a batch size of 100. These tasks were run on a NVIDIA K20m GPU with CUDA 9.0 using the Chainer v4.0 (Tokui et al. (2015)) as the framework. For the ImageNet datasets, VGG16 (Simonyan & Zisserman (2014)) was used as the network architecture, and the network was trained using stochastic gradient descent with momentum as the optimization method with a batch size of 100 samples. The initial learning rate of 0.01 was exponentially decayed by multiplying it by 0.95 at each epoch. Weight decay regularization was used with a decay rate of $5 \cdot 10^{-4}$ by following the original paper. In the VGG16 architecture, dropout was applied for the first two fully connected layers with 50% as dropout ratio. A NVIDIA V100 GPU with CUDA 10.0 was used for the training with ImageNet datasets. For all datasets, data augmentation was used by extracting a patch from a random position of the input image and by performing random horizontal flipping during training (Krizhevsky et al. (2012)). For the validation set, the patch from the center position was extracted and fed into the classifier without horizontal flipping.

## 3.2 IMPROVEMENTS BY MULTI-SAMPLE DROPOUT

Figure 2 plots the trends in training losses, training errors and validation errors against training time for three configurations: trained with the original dropout, with multi-sample dropout, and without dropout. For multi-sample dropout, the losses for eight dropout samples were averaged. How the number of dropout samples affects performance is discussed in the next section. The figure shows that multi-sample dropout achieved faster convergence than the original dropout for all datasets (including SVHN, which is not shown here due to space limitation), i.e. both losses and errors became smaller with the same training time for both training sets and validation sets. As is common in regularization techniques, dropout achieves better generalization (i.e., lower validation error rates) compared with the "no dropout" case at the expense of slower training. Multi-sample dropout alleviates this slowdown while still achieving better generalization.

Table 1 summarizes the final training losses, training error rates, and validation error rates. For the CIFAR-10, CIFAR-100, and SVHN datasets, the results for 200 epochs were averaged after training the network for 1,800 epochs. For ImageNet-100, we trained VGG16 for 16 million images. For ImageNet, we trained VGG16 for 96 million images. After training, the networks trained with multi-sample dropout were observed to have reduced losses and error rates for all datasets compared with those of the original dropout. The original dropout increased the training losses and error rates by avoiding overfitting for all datasets compared with the no dropout case. Multi-sample dropout achieved lower training losses than no dropout for some datasets while avoiding overfitting.

## 3.3 EFFECTS OF PARAMETERS ON PERFORMANCE

**Number of dropout samples:** Figure 3(a) and 3(b) compare the training losses and validation errors for different numbers of dropout samples for CIFAR-100 against the number of training epochs. Using a larger number of dropout samples made convergence faster. With eight dropout samples, for example, the validation accuracy reached 95% of the final accuracy after the 25th epoch while it was after the 42nd epoch with the original dropout. A clear relationship is evident between the number of dropout samples and the speedup of convergence in terms of the number of iterations (epochs) with up to 64 dropout samples. For the validation error shown in Figure 3(b), the benefit of using more than eight dropout samples was not significant.

The accelerated training in terms of the number of iterations (epochs) due to using more dropout samples came at cost: increased execution time per iteration. The execution time per iteration relative to that of original dropout is shown in Table 2 for different numbers of dropout samples. The VGG16 network architecture was used for the ImageNet dataset and a smaller 8-layer CNN was used for

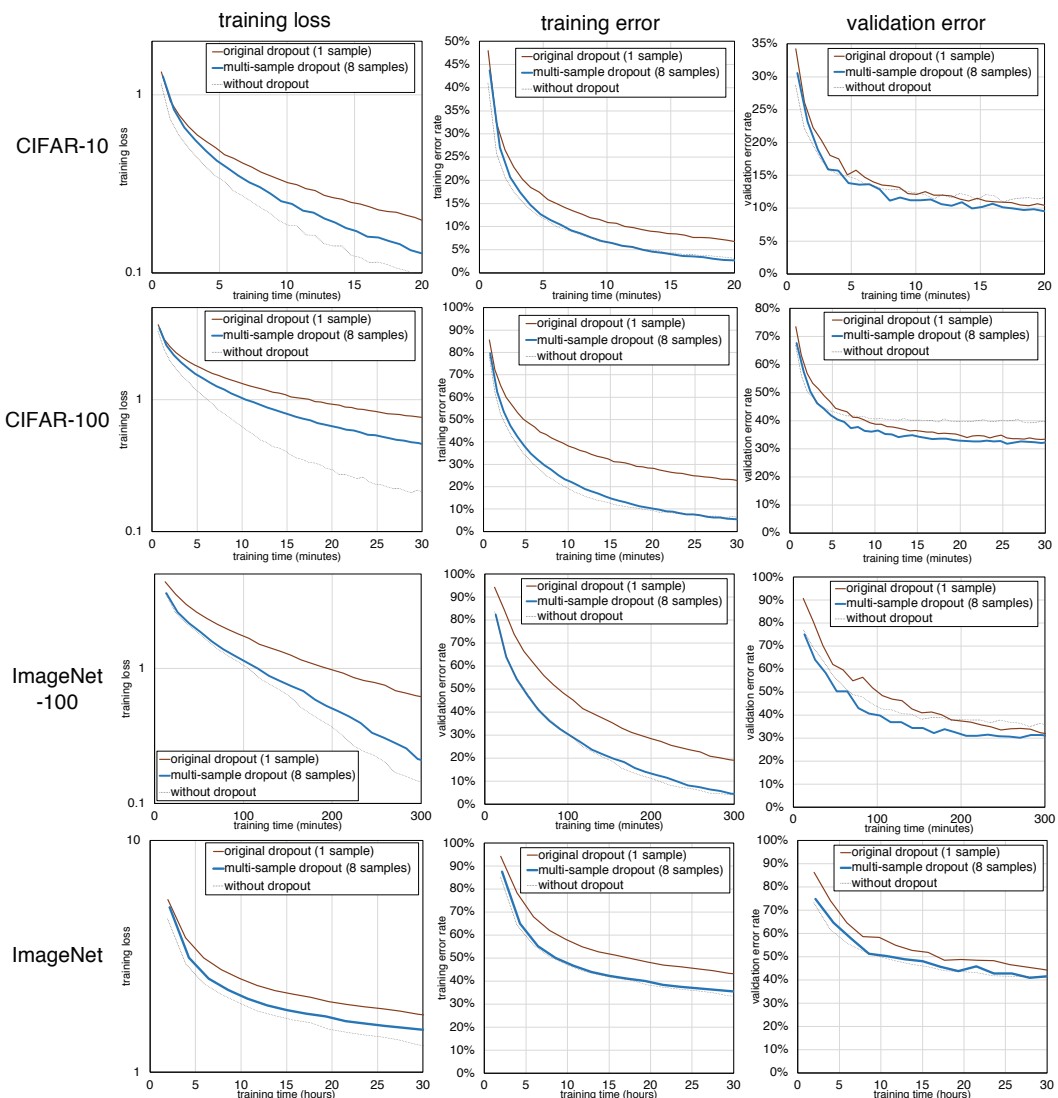

Figure 2: Trends in training losses, training errors and validation errors against training time for original dropout and multi-sample dropout. Multi-sample dropout achieved faster convergence than the original dropout. Note that training losses and errors for multi-sample dropout include the effects of the inherent ensemble mechanism, but no ensemble was used while evaluating validation sets.

Table 1: Training losses, training error rates, and validation error rates with original dropout and multi-sample dropout (after trained for 96 million images for ImageNet, 16 million images for ImageNet-100, and 1,800 epochs for others). Multi-sample dropout reduced losses and error rates compared with original dropout.

| dataset | training loss | | | training error rate | | | validation error rate | | |
|---|---|---|---|---|---|---|---|---|---|
| | original dropout | multi-sample dropout | no dropout | original dropout | multi-sample dropout | no dropout | original dropout | multi-sample dropout | no dropout |
| CIFAR-10 | 0.0170 | **0.0045** | 0.0074 | 0.39% | **0.01%** | 0.11% | 8.05% | **7.46%** | 9.13% |
| CIFAR-100 | 0.1530 | 0.0537 | **0.0325** | 4.49% | **0.04%** | 0.40% | 30.0% | **29.3%** | 36.6% |
| SVHN | 0.0188 | **0.0046** | 0.0062 | 0.45% | **0.01%** | 0.09% | 4.46% | **4.21%** | 4.82% |
| ImageNet-100 | 0.0472 | **0.0344** | 0.0346 | 1.41% | **0.89%** | 0.92% | 27.1% | **25.9%** | 31.4% |
| ImageNet | 0.4922 | 0.3808 | **0.1777** | 14.4% | 8.61% | **3.88%** | 30.5% | **30.1%** | 30.8% |

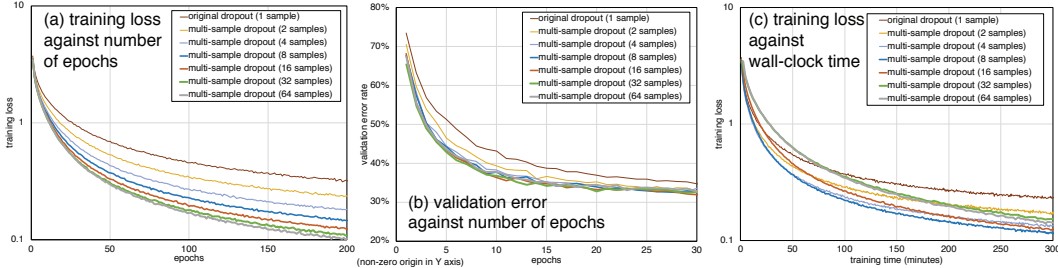

Figure 3: Training losses and validation errors during training for different numbers of dropout samples. Using more dropout samples makes convergence faster in terms of number of iterations at the cost of increased execution time per iteration.

Table 2: Execution time per iteration relative to that of original dropout for different numbers of dropout samples. Increasing the number of dropout samples lengthened the computation time per iteration. We failed to execute VGG16 with 16 dropout samples due to the out-of-memory error.

| dataset | network | original dropout | multi-sample dropout | | | | | |
|---|---|---|---|---|---|---|---|---|
| | | 1 sample | 2 samples | 4 samples | 8 samples | 16 samples | 32 samples | 64 samples |
| CIFAR-10 | | 1.00 | 1.04 | 1.08 | 1.19 | 1.44 | 2.16 | 3.90 |
| CIFAR-100 | 8-layer CNN | 1.00 | 1.04 | 1.09 | 1.19 | 1.44 | 2.22 | 3.55 |
| SVHN | | 1.00 | 1.04 | 1.08 | 1.19 | 1.47 | 2.07 | 3.82 |
| ImageNet | VGG16 | 1.00 | 1.01 | 1.05 | 1.09 | - | - | - |

the other datasets, as mentioned above. Because a larger network tends to spend more time in deep convolutional layers than in the fully connected layers, which are duplicated in our multi-sample dropout technique, the overhead in execution time compared with that of the original dropout is more significant for the smaller network than it is with the VGG16 network architecture. Multi-sample dropout with eight dropout samples increased the execution time per iteration by about 9% for the VGG16 architecture and by about 19% for the small network. For even smaller networks, the increases in the execution time per iteration became too significant and hence using multi-sample dropout may not make sense as discussed in Appendix using multilayer perceptron as an example.

Consideration of the increased execution time per iteration along with the reduced number of iterations revealed that multi-sample dropout achieves the largest speed up in training time when a moderate number of dropout samples, such as 8, is used, as shown in Figure 3(c). Using an excessive number of dropout samples may actually slow down the training.

The final values of the training losses, training error rates, and validation error rates are shown in Figure 4. The average losses and error rates between the 1,800th and 2,000th epoch were used. Multi-sample dropout achieved lower losses and error rates as the number of dropout samples was increased. The gains were not significant when the number was increased above eight.

From these observations, it was determined that eight is a reasonable value for the number of dropout samples, and it is used in other experiments.

**Dropout ratio:** Another important parameter is the dropout ratio. In the 8-layer CNN used for CIFAR datasets, 40% and 30% were used as the ratios in the two dropout layers. These values were tuned for the original dropout. Here it is shown how multi-sample dropout works for the CIFAR-10 dataset with various dropout ratios: {10%, 10%}, {25%, 25%}, {40%, 30%} (default), {50%, 50%}, {75%, 75%}, {90%, 90%}.

Figure 5(a) shows the final validation error rates. Regardless of the dropout ratio setting, multi-sample dropout outperformed the original dropout. Figure 5(b) compares the trend in convergence of training losses for dropout ratios of 25% and 75%. For both configurations, the acceleration in training over the original dropout was similar to that for the default case (Figure 2). For the original dropout with a 75% dropout ratio, there were spikes in the training loss history, which were not observed with

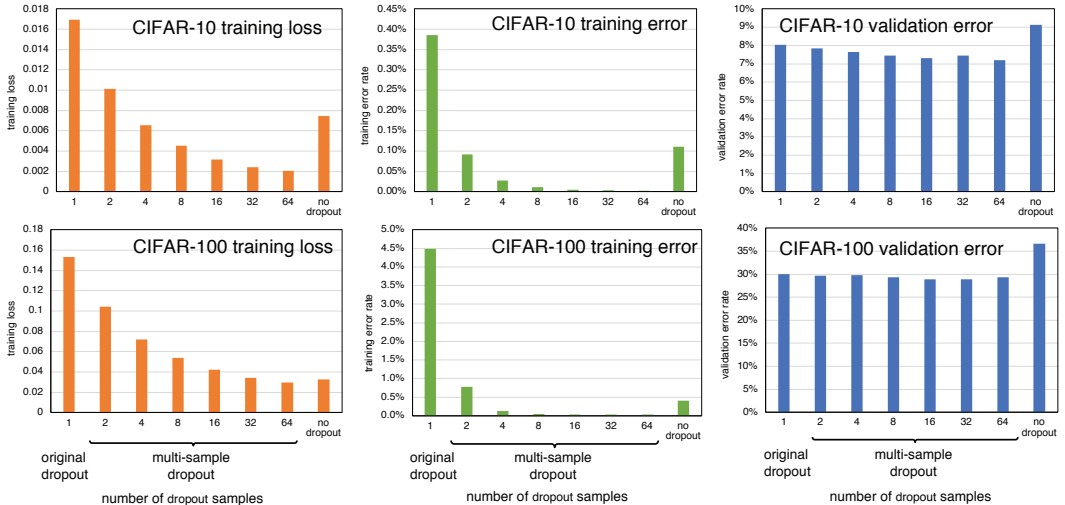

Figure 4: Losses and error rates after the training for different numbers of dropout samples. Multi-sample dropout achieved lower losses and error rates with more dropout samples.

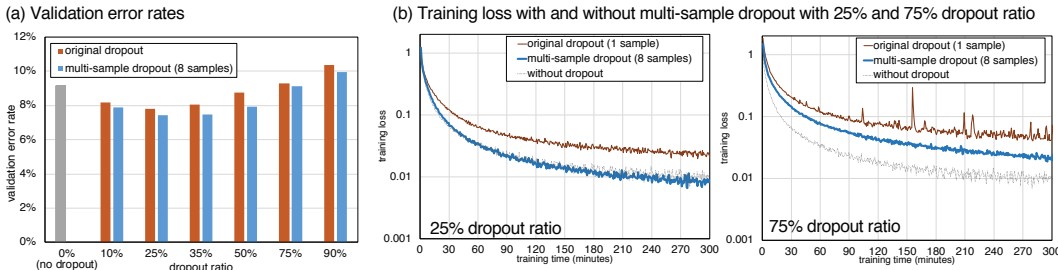

Figure 5: (a) Validation error rates and (b) trend in training losses with original and multi-sample dropout for various drop ratios. Here, 35% dropout ratio means two dropout layers use 40% and 30% respectively. Multi-sample dropout works regardless of the dropout ratio.

smaller dropout ratios. Multi-sample dropout did not cause such spikes and exhibited a smoother history and a stable training by reducing gradient noise.

These results show that multi-sample dropout does not depend on a specific dropout ratio to achieve improvements and that it can be used with a wide range of dropout ratio settings.

### 3.4 WHY MULTI-SAMPLE DROPOUT IS EFFICIENT

As discussed in Section 2.2, the effect of multi-sample dropout with $M$ dropout samples is similar to that of enlarging the size of a minibatch $M$ times by duplicating each sample in the minibatch $M$ times. This is the primary reason for the accelerated training with multi-sample dropout. This is illustrated in Figure 6: the training losses with multi-sample dropout match well with those of the original dropout using duplicated data in a minibatch. We also observed that the original dropout with duplicated data achieved comparable validation error rates to multi-sample dropout. If the same sample is included in a minibatch multiple times, the results from the multiple samples are ensembled when the parameters are updated, even if there is no explicit ensembling in the network. Duplicating a sample in input and ensembling them at the parameter updates seems to have a quite similar effect on training to that of multi-sample dropout, which duplicates a sample at the dropout and ensembles them at the end of the forward pass. However, duplicating data $M$ times make the execution time per iteration (and hence the total training time) $M$ times longer than without duplication. Multi-sample dropout achieves a similar speedup at a much smaller computation cost.

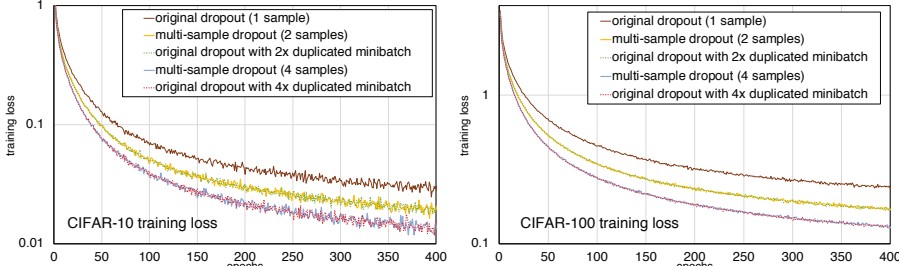

Figure 6: Comparison of original dropout with data duplication in minibatch and multi-sample dropout. X-axis shows number of epochs. For fair comparison, we do not use additional diversity due to horizontal flipping and zero padding for multi-sample dropout in this experiment.

## 4 RELATED WORK

The multi-sample dropout regularization technique presented in this paper can achieve better generalization and faster training than the original dropout. Dropout is one of the most widely used regularization techniques, but a wide variety of other regularization techniques for better generalization have been reported. They include, for example, weight decay (Krogh & Hertz (1991)), data augmentation (Cubuk et al. (2018), Chawla et al. (2002), Zhang et al. (2017), Inoue (2018)), label smoothing (Szegedy et al. (2016)), and batch normalization (Ioffe & Szegedy (2015)). Although batch normalization is aimed at accelerating training, it also improves generalization. Many of these techniques are network independent while others, such as Shake-Shake (Gastaldi (2017)) and Drop-Path (Larsson et al. (2017)), are specialized for a specific network architecture.

The success of dropout led to the development of many variations that extend the basic idea of dropout (e.g. Ghiasi et al. (2018), Huang et al. (2016), Tompson et al. (2014), DeVries & Taylor (2017)). The techniques reported use a variety of ways to randomly drop information in the network. For example, DropConnect (Wan et al. (2013)) discards randomly selected connections between neurons. DropBlock (Ghiasi et al. (2018)) randomly discards areas in convolution layers while dropout is typically used in fully connected layers after the convolution layers. Stochastic Depth (Huang et al. (2016)) randomly skip layers in a very deep network. However, none of these techniques use the approach used in our multi-sample dropout. Also many of them can be used with multi sampling technique to make the divergence among dropout samples.

Multi-sample dropout calculates the final prediction and loss by averaging the results from multiple loss functions. Several network architectures have multiple exits with loss functions. For example, GoogLeNet (Szegedy et al. (2015)) has two early exits in addition to the main exit, and the final prediction is made using a weighted average of the outputs from these three loss functions. Unlike multi-sample dropout, GoogLeNet creates the two additional exits at earlier positions in the network. Multi-sample dropout creates multiple uniform exits, each with a loss function, by duplicating a part of the network.

## 5 CONCLUSION

In this paper, we described multi-sample dropout, a regularization technique for accelerating training and improving generalization. The key is creating multiple dropout samples at the dropout layer while the original dropout creates only one sample. Multi-sample dropout can be easily implemented using existing deep learning frameworks by duplicating a part of the network after the dropout layer. Experimental results using image classification tasks demonstrated that multi-sample dropout reduces training time and improves accuracy. Because of its simplicity, the basic idea of the multi-sampling technique can be used in wide range of neural network applications and tasks.

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

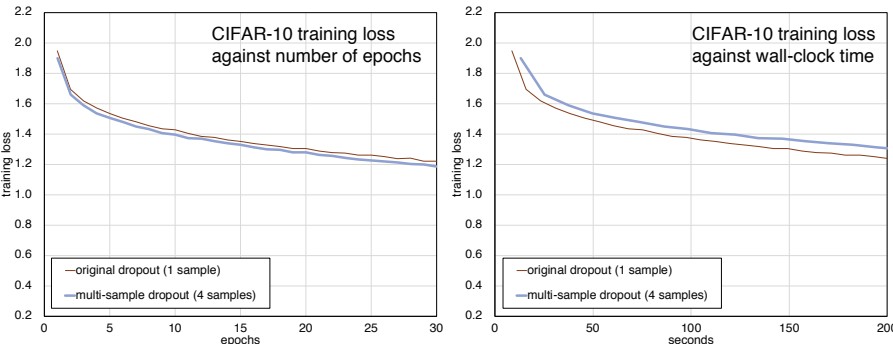

Figure 7: Comparison of training loss using the multilayer perceptron with and without multi-sample dropout for CIFAR-10.

## A  APPENDIX

### A.1  APPLYING MULTI-SAMPLE DROPOUT FOR SHALLOW NETWORKS

As discussed in Section 2.1, multi-sample dropout is mainly targeting deep convolutional neural networks in which most of the computation time is consumed in the convolution layers before the dropout. Here, we show the effect of multi-sample dropout on shallow networks using a multilayer perceptron as an example. We use a network consists of four fully connected layers each has 2,000 neurons. We apply dropout for each fully connected layer. For multi-sampling, we created four dropout samples at the last fully connected layer; i.e. only one layer is duplicated. Figure 7 shows the training loss for CIFAR-10 dataset with and without multi-sample dropout. Multi-sample dropout yields smaller training loss compared to the original dropout after training the same number of epochs. However, due to the increase in the computation time per iteration, multi-sample dropout actually degraded the training speed in terms of the training time; the execution time per iteration increased by more than 50% even we created only four dropout samples. Hence, to make multi-sample dropout effective, it is important that dropout is typilly applied to layers near the end of the network to limit the number of operations duplicated for multiple dropout samples.

### A.2  IMPROVEMENTS FROM OTHER SOURCES OF DIVERSITY AMONG DROPOUT SAMPLES

As discussed in Section 2.3, zero padding at a pooling layer and horizontal flipping can be used as sources of diversity among dropout samples in addition to the different masks in dropout.

In the current implementation, as detailed in the next section using pseudocode, the horizontal flipping is applied immediately before the first fully connected layer; a half of the dropout samples are horizontal flipped deterministically. When pooling an image with a size that is not a multiple of the window size, e.g., when applying 2x2 pooling to a 7x7 image, the zero padding can be added on the left or right and at the top or bottom. Here, zero padding is added on the right (and bottom) for half of the dropout samples and on the left (and bottom) for the other half. The location of the zero padding is controlled by using horizontal flipping; i.e., "flip, pool with zero padding on right, and then flip" is equivalent to "pool with zero padding on left."

Figure 8 shows how these additional sources contribute to training acceleration. Although their contributions to the training speed are not negligible, they are much smaller than those of dropout. Nevertheless, these results show that the multi-sampling technique can work with not only dropout but also with other sources of divergence among samples while the dropout is an ideal source of divergence among multiple samples.

### A.3  DETAIL OF THE USED NETWORK ARCHITECTURE

The network design of the 8-layer CNN used for CIFAR and SVHN datasets are shown . We also present an overview of the implementation using Chainer for more detail.

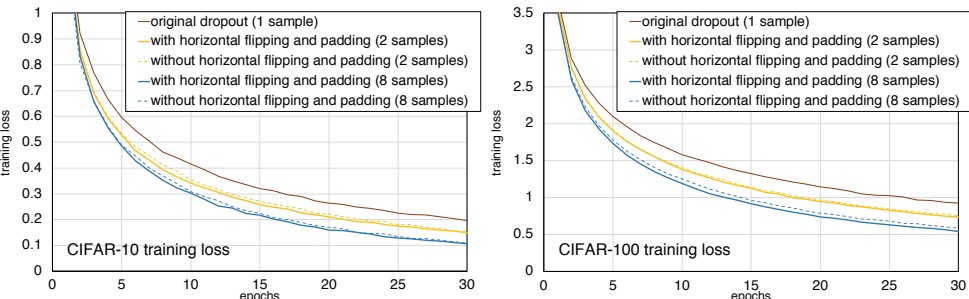

Figure 8: Comparison of training loss with and without horizontal flipping to create additional diversity among dropout samples in multi-sample dropout. X-axis shows number of epochs.

```
                                     tensor size
 Batch Normalization                 28x28x3
 Convolution 3x3 & ReLU              28x28x64
 Batch Normalization
 Convolution 3x3 & ReLU              28x28x96
 Max Pooling 2x2                     14x14x96
 Batch Normalization
 Convolution 3x3 & ReLU              14x14x96
 Batch Normalization
 Convolution 3x3 & ReLU              14x14x128
 Max Pooling 2x2                     7x7x128
 Batch Normalization
 Convolution 3x3 & ReLU              7x7x128
 Batch Normalization
 Convolution 3x3 & ReLU              7x7x192
 for i = 0 to numSamples/2-1:
   if (i & 1) != 0: HorizontalFlip
   Max Pooling 2x2                   4x4x192
   Batch Normalization
   for j = 0 to 1:
     Dropout 40% dropout ratio
     if j == 1: HorizontalFlip
     Fully Connected & ReLU          512
     Dropout 30% dropout ratio
     Fully Connected                 10 or 100
     Softmax & CrossEntropy
 Average all losses
```

Figure 9: Design of classification network (with eight weight layers) with multi-sample dropout used for CIFAR and SVHN. Refer Section 2.3 for details of horizontal flipping used to create more diversity among dropout samples in addition to dropout.

```
class MultiSampleDropoutCNN(chainer.Chain):
    def __init__(self, nlabels):
        super(MultiSampleDropoutCNN, self).__init__(
        l1 = L.BatchNormalization(3),
        l2 = L.Convolution2D(3, 64, 3, pad=1),
        l4 = L.BatchNormalization(64),
        l5 = L.Convolution2D(None, 96, 3, pad=1),
        l8 = L.BatchNormalization(96),
        l9 = L.Convolution2D(None, 96, 3, pad=1),
        l11 = L.BatchNormalization(96),
        l12 = L.Convolution2D(None, 128, 3, pad=1),
        l15 = L.BatchNormalization(128),
        l16 = L.Convolution2D(None, 128, 3, pad=1),
        l18 = L.BatchNormalization(128),
        l19 = L.Convolution2D(None, 192, 3, pad=1),
        l22 = L.BatchNormalization(192),
        l24 = L.Linear(None, 512),
        l27 = L.Linear(None, nlabels),
        )

    def __call__(self, h0, t):
        with chainer.using_config('train', self.train):
            h1 = self.l1(h0)  # batch normalization
            h2 = self.l2(h1)  # convolutional
            h3 = F.relu(h2)
            h4 = self.l4(h3)  # batch normalization
            h5 = self.l5(h4)  # convolutional
            h6 = F.relu(h5)
            h7 = F.max_pooling_2d(h6, 2)
            h8 = self.l8(h7)  # batch normalization
            h9 = self.l9(h8)  # convolutional
            h10 = F.relu(h9)
            h11 = self.l11(h10)  # batch normalization
            h12 = self.l12(h11)  # convolutional
            h13 = F.relu(h12)
            h14 = F.max_pooling_2d(h13, 2)
            h15 = self.l15(h14)  # batch normalization
            h16 = self.l16(h15)  # convolutional
            h17 = F.relu(h16)
            h18 = self.l18(h17)  # batch normalization
            h19 = self.l19(h18)  # convolutional
            h20 = F.relu(h19)
            h21 = F.max_pooling_2d(h20, 2)
            h22 = self.l22(h21)  # batch normalization
            # dropout sample 1
            h23a = F.dropout(h22, ratio = 0.40)
            h24a = self.l24(h23a)  # fully connected
            h25a = F.relu(h24a)
            h26a = F.dropout(h25a, ratio = 0.30)
            h27a = self.l27(h26a)  # fully connected

            if self.enableMultiSampleDropout:
                # dropout sample 2
                h23b = F.dropout(h22, ratio = 0.40)
                h24b = self.l24(F.flip(h23b, axis=3))  # fully connected after horizontal flip
                h25b = F.relu(h24b)
                h26b = F.dropout(h25b, ratio = 0.30)
                h27b = self.l27(h26b)  # fully connected

                # dropout sample 3
                h21c = F.max_pooling_2d(F.flip(h20, axis=3), 2)  # pooling after horizontal flip
                h22c = self.l22(h21c)  # batch normalization
                h23c = F.dropout(h22c, ratio = 0.40)
                ...

                # dropout sample 4
                h23d = F.dropout(h22c, ratio = 0.40)
                h24d = self.l24(F.flip(h23d, axis=3))  # fully connected after horizontal flip
                ...

                self.var_loss = (F.softmax_cross_entropy(h27a, t) + ....) / 8.0
                self.y = (h27a + h27b + h27c ....) / 8.0  # for making the prediction
                # here, no gain observed with
                # self.y = (F.softmax(h27a) + F.softmax(h27b) +...) / 8.0
            else:
                self.var_loss = F.softmax_cross_entropy(h27a, t)
                self.y = h27a

        return self.var_loss
```

Figure 10: Example of an implementation of CNN with multi-sample dropout shown in Figure 8 on Chainer using eight dropout samples

