# OpenReview forum: "Multi-Sample Dropout for Accelerated Training and Better Generalization"
_ICLR.cc/2020/Conference — Reject_

### Official Review · AnonReviewer1 · 2019-10-22
**Official Blind Review #1**

**Rating:** 1

**Review:**

This paper propose an ensemble of dropout: it applies multiple copies of a neural net with different dropout configurations (i.e., dropout masks) to the same mini-batch, and the training loss is computed as the sum of losses incurred on the multiple copies. They claim that such ensembling can improve training performance over the original dropout without increasing too much computation. Experiments on several datasets show that the proposed method can achieve slightly better validation accuracies than the original dropout.

Main comments and questions:

1) The contribution of this paper is very limited: it simply applies the ensemble method to dropout (ensemble method can be applied to any existing models). Moreover, this contribution might not be very necessary: dropout is itself an ensemble method, so an ensemble of dropout does not make much sense.

2) The paper keeps claiming that the computation will not dramatically increase because only the computations after dropout neede to be re-computed if applying multiple dropout masks to the same data(batch). This is only true for the case when dropout is used in the last few layers. For a rich class of modern neural nets such as WideResNet and many Language models, this is not true since each block contains dropout and the model is a stack of many blocks. Hence, the advantage claimed here can be only applied to very limited cases. In addition, only the layers after dropout can get the benefits of dropout training since only they get different inputs for different dropout masks but need to produce similar final outputs.

In Sec 2.2, the authors said their method "can enjoy similar gains without a huge increase in computation cost because it duplicates only the operations after dropout". So the computational advantage does not hold if we apply dropout in earlier/shallow layers. For example, if you apply dropout in the first layer, the method needs to execute all the operations after the first layer for each dropout sample (dropout changes everything after it), hence it increases the computation time nearly M times (for M dropout-samples) and loses the computational advantage. This might be the reason why the experiments only consider DNNs with dropout in the last fully-connected layers.

3) The experiments did not evaluate the method on any modern neural nets and only tried small neural nets with dropout only applied to the last one or two fully-connected layers. It is not clear how it performs when used to train the modern models, which might prefer dropout above the last few layers.

4) The dropout with 1-sample in experiments is not a standard baseline in any previous works. The two standard ways to do inference on dropout trained model are 1) one-time inference without any dropout, or 2) ensemble the inference results of multiple dropout samples. One dropout sample suffers from a higher variance. Hence, the comparison is unfair.

5) It is not professional to report only training loss and validation error for some experiments, but report training loss/error and validation error for some others. In fact, it does not make sense to compare training loss/error of different dropout methods, since dropout always reduces training loss but improve generalization.

6) Except for the vanilla dropout, no other baseline is compared. The paper simply ignores many successful variants of dropout proposed in the past several years.

----------------

Update:

The rebuttal does not answer my questions. So I will keep my rating unchanged.

**Experience Assessment:**

I have published one or two papers in this area.

**Review Assessment: Checking Correctness Of Derivations And Theory:**

I carefully checked the derivations and theory.

**Review Assessment: Checking Correctness Of Experiments:**

I carefully checked the experiments.

**Review Assessment: Thoroughness In Paper Reading:**

I read the paper thoroughly.

---

> ### Author Response · Authors · 2019-11-14
> **author response to Review #1**
>
> Thank you so much for the detailed comments.
>
> Overall, I like to improve the explanation to make the descriptions clearer.
>
> > The dropout with 1-sample in experiments is not a standard baseline in any previous works
> At the inference time, we do not apply dropout for both standard and multi-sample dropout evaluations.

---

### Official Review · AnonReviewer2 · 2019-10-23
**Official Blind Review #2**

**Rating:** 3

**Review:**

The paper proposes a new variation of Dropout -- Multi-Sample Dropout. The method is said to 1) accelerate training, and 2) decrease validation error. To achieve that, the authors propose to average a loss function over several dropout samples per object during training. This leads to faster convergence in terms of iterations, though at the cost of multiple forward-passes. In order to decrease the computational time of one iteration, the authors propose to evaluate the part of a network before the first dropout layer only ones and duplicate the remaining part with different dropout masks and shared weights. This decreases the run time of the naive approach and leads to faster convergence in terms of time in comparison to the original dropout. The authors also show that the model trained with the proposed method achieves lower validation error.

The only confusing part of the method for me is the prediction. In the second paragraph in section 2.1, the authors state that during inference neurons are not discarded, and only one dropout sample is used for the prediction. This is confusing because using a dropout sample implies the dropping of neurons. Could you please elaborate on this? Moreover, in the first paragraph of section 2.1, the authors state the prediction is based on the average of outputs from the last fully-connected layer, which is different from the inference procedure. Do you use different prediction methods to evaluate the performance of the train and test sets?

There are several concerns regarding empirical evaluation and baselines:
(1) The improvements in validation error shown in Tab.1 seem to be insignificant, and confidence intervals are needed for justification.
(2) To the best of knowledge, there is at least one more paper that proposes a technique for accelerating dropout training --- Fast Dropout [http://proceedings.mlr.press/v28/wang13a.html]. The method seems to be a direct competitor and should be considered as a baseline and be included in the relevant work section with further discussion.

Overall, the proposed approach is heuristic, and the novelty is very limited. Along with empirical evaluation issues, I would suggest improving the work and rejecting the current version.

Additional comments:
1. I think it would be beneficial to discuss other works that similarly interpret dropout as an ensembling technique: Dropout as a Bayesian Approximation: Representing Model Uncertainty in Deep Learning [https://arxiv.org/abs/1506.02142] and Variational Dropout and the Local Reparameterization Trick [https://arxiv.org/abs/1506.02557].

**Experience Assessment:**

I have published in this field for several years.

**Review Assessment: Checking Correctness Of Derivations And Theory:**

N/A

**Review Assessment: Checking Correctness Of Experiments:**

I assessed the sensibility of the experiments.

**Review Assessment: Thoroughness In Paper Reading:**

I read the paper thoroughly.

---

> ### Author Response · Authors · 2019-11-14
> **author response to review #2**
>
> I greatly thank for the valuable comments.
>
> > Fast Dropout and other works
> Thank you for pointing this out. We are going to discus them in future versions.
>
> > confidence intervals are needed for justification
> We are still improving the experiments by adding more runs as you suggested.

---

### Official Review · AnonReviewer3 · 2019-10-26
**Official Blind Review #3**

**Rating:** 1

**Review:**

PAPER SUMMARY: The paper proposes a new and efficient implementation of dropout in which multiple dropout samples are obtained from a single input during training. The authors claim that this enhanced dropout technique (i) accelerates training and (ii) improves generalization by achieving lower error rates than standard dropout in training and validation sets. Experiments in image classification tasks for 4 different datasets (CIFAR-10, CIFAR-100, IMAGENET and SVHN) are presented and the effect of the number of dropout samples and dropout ratio are investigated.

REVIEW SUMMARY: Despite the clear computational advantages of the proposed implementation, this paper should be rejected because  (1) the authors claim multi-sample dropout to be a “new regularization technique”, but the conceptual differences between standard dropout and their proposed method are not clearly stated, (2) one of the main claims regarding “better generalization” is not well supported by their experiments, and (3) the experiments do not consider the stochasticity of the methods they are evaluating.

DETAILED COMMENTS: The multiple-sample dropout proposed by the authors leverages a parallel structure to provide significant acceleration in training, which is supported by experimental results. However, their claims regarding the introduction of a “new regularization technique” and “better generalization” are not convincingly proven in the paper. As mentioned in the paper, the proposed technique of multiple-sample dropout is analogous to the original implementation of dropout, but with larger batch size and duplicated samples. Therefore, the advantage of the proposed method is simply computational efficiency, but not a new “regularization technique”. In addition, while the paper claims the method provides better generalization, the results shown in Figure 6, in which the training losses for their multi-sample dropout and original dropout with duplicated samples are not significantly different, suggest that their validation error might not be significantly different either. The experiments shown in Figure 2, from which they conclude that their method generalizes better, are not conclusive since the validation error is evaluated at a stage in which the original dropout might have not converged, given the slower training. Therefore, the observed differences in validation error might come from the differences in training speed, and not from a better generalization ability of the model. Moreover, in the experiments the authors present single realizations of each training scenario, disregarding the stochastic nature of dropout-based techniques. In particular, the validation error curves observed in Figure 3 (where they analyze the effect of the number of dropout samples) seem very noisy, and the differences observed might simply come from the stochasticity of the training process. The authors should have executed multiple training runs with the same set of hyperparameters, and reported the statistics of their performance (same for results presented in Figure 2 and Table 1).  Overall, the advantages regarding training acceleration and computational efficiency are clear and well supported by the experiments. However, the novelty of the proposed technique and their claim regarding better generalization is not well supported theoretically or experimentally.

For the experiments, the authors should consider:
1)	Report validation errors after convergence of both methods
2)	Run the training scenarios multiple times and report performance statistics instead of single run performance
3)	Report validation curves for experiments in section 3.4 (Figure 6).

**Experience Assessment:**

I have published one or two papers in this area.

**Review Assessment: Checking Correctness Of Derivations And Theory:**

N/A

**Review Assessment: Checking Correctness Of Experiments:**

I carefully checked the experiments.

**Review Assessment: Thoroughness In Paper Reading:**

I read the paper thoroughly.

---

> ### Author Response · Authors · 2019-11-14
> **author response for review #3**
>
> Thank you so much for the valuable advice.
>
> > but the conceptual differences between standard dropout and their proposed method are not clearly stated
> The key point of out method is creating multiple dropout samples and averaging the loss for them as the final loss during the training. Using each training example multiple times with the standard dropout makes the execution time longer, while the increase in the execution time with our method is much smaller since we duplicate only a part of the network after the dropout.
>
> > not well supported by their experiments
> We are still improving the experiments by adding more runs as you suggested.
>
> > Report validation errors after convergence of both methods
> Table 1 and Fig 4 show the validation errors after convergence.

---

### Decision · Program_Chairs · 2019-12-19

**Decision:**

Reject

**Comment:**

This paper proposes a multi-sample variant of dropout, claiming that it accelerates training and improves generalization. CIFAR10/100, ImageNet and SVHN results are presented, along with a few ablations.

Reviewers were in agreement that the novelty of the contribution appears to be very limited, the evidence for the claims is not strong, and that the applicability of the method for achieving efficiency gains is limited to architectures that only apply dropout very late in processing, precluding applicability to models that employ dropout throughout. Importantly, comparisons to Fast Dropout (Wang 2013) seem highly relevant and are missing.

While the reviewers acknowledged some of the criticisms, virtually no arguments were offered to rebut them and no updates were made to address them. I therefore recommend rejection.